# COVID-19 and Parasitic Co-Infection: A Hypothetical Link to Pulmonary Vascular Disease

**DOI:** 10.3390/idr17020019

**Published:** 2025-02-27

**Authors:** Peter S. Nyasulu, Jacques L. Tamuzi, Rudolf K. F. Oliveira, Suellen D. Oliveira, Nicola Petrosillo, Vinicio de Jesus Perez, Navneet Dhillon, Ghazwan Butrous

**Affiliations:** Division of Epidemiology & Biostatistics, Department of Global Health, Faculty of Medicine & Health Sciences, Stellenbosch University, France Van Zijl Drive, Tygerberg, Cape Town 7505, South Africa; drjacques.tamuzi@gmail.com (J.L.T.); rudolf.oliveira@unifesp.br (R.K.F.O.); suelleno@uic.edu (S.D.O.); n.petrosillo@policlinicocampus.it (N.P.); vdejesus@stanford.edu (V.d.J.P.); ndhillon@kumc.edu (N.D.); gbutrous@gmail.com (G.B.)

**Keywords:** co-infections, COVID-19, long COVID, PVD, pulmonary vascular remodelling and hypothesis

## Abstract

**Background/Objectives**: Before the Coronavirus disease 2019 (COVID-19) era, the global prevalence of pulmonary arterial hypertension (PAH) was between 0.4 and 1.4 per 100,000 people. The long-term effects of protracted COVID-19 associated with pulmonary vascular disease (PVD) risk factors may increase this prevalence. According to preliminary data, the exact prevalence of early estimates places the prevalence of PVD in patients with severe acute respiratory syndrome coronavirus 2 (SARS-CoV-2) infection at 22%, although its predictive value remains unknown. PVD caused by COVID-19 co-infections is understudied and underreported, and its future impact is unclear. However, due to COVID-19/co-infection pathophysiological effects on pulmonary vascularization, PVD mortality and morbidity may impose a genuine concern—both now and in the near future. Based on reported studies, this literature review focused on the potential link between COVID-19, parasitic co-infection, and PVD. This review article also highlights hypothetical pathophysiological mechanisms between COVID-19 and parasitic co-infection that could trigger PVD. **Methods**: We conducted a systematic literature review (SLR) searching peer-reviewed articles, including link between COVID-19, parasitic co-infection, and PVD. **Results**: This review hypothesized that multiple pathways associated with pathogens such as underlying schistosomiasis, human immunodeficiency virus (HIV), pulmonary tuberculosis (PTB), pulmonary aspergillosis, *Wuchereria bancrofti*, *Clonorchis sinensis*, paracoccidioidomycosis, human herpesvirus 8, and scrub typhus coupled with acute or long COVID-19, may increase the burden of PVD and worsen its mortality in the future. **Conclusions**: Further experimental studies are also needed to determine pathophysiological pathways between PVD and a history of COVID-19/co-infections.

## 1. Introduction

Pulmonary vascular diseases (PVDs), due to infectious causes, are increasing in prevalence, especially in low- and middle-income nations (LMICs), where the disease burden is higher than in high-income countries [1]. Developed nations account for about 0.05% of PVDs worldwide, while underdeveloped nations account for about 17% of PVDs linked to infectious illnesses (mostly schistosomiasis and HIV) [1,2]. In the developing world, a wide range of infectious diseases can cause PVDs, resulting predominantly in pulmonary arterial hypertension (PAH) [2]. PVDs are progressive, severe, and hemodynamic disorders that may cause high mortality if not well treated [3]. Increased pulmonary vascular resistance (PVR) is caused by diseases of the pulmonary vasculature in pulmonary embolism, chronic thromboembolic pulmonary hypertension (CTEPH), and PAH [4]. Since the World Health Organization (WHO) conducted the first World Symposium on Pulmonary Hypertension (WSPH) in Geneva in 1973, PAH has been defined as mean pulmonary arterial pressure (mPAP) ≥ 25 mmHg, as assessed by right heart catheterisation (RHC) in the supine position at rest [5]. At rest, the mPAP is 14.0 ± 3.3 mmHg; this value is independent of gender and ethnicity and may be only slightly altered by age and posture [6]. PAH prevalence ranged from 0.4 to 1.4 per 100,000 people worldwide [7]. PAH prevalence is projected to rise following the COVID-19 event due to the long-term impacts of protracted COVID related to PVD risk factors such as obesity, cardiovascular disease, and various infectious diseases (schistosomiasis, HIV, pulmonary tuberculosis...). PAH may rise more than expected due to COVID-19-associated pulmonary hemodynamic changes via lung vascular endothelial dysfunction, vascular inflammation, increased vascular permeability, and thrombotic microangiopathy mechanisms similar to those contributing to other PVDs [8,9]. Lung parenchymal damage and altered pulmonary haemodynamic may determine PAH and secondary right ventricular (RV) involvement in patients with COVID-19, even in non-advanced disease stages [10].

There is a major concern regarding the occurrence and persistence of pulmonary vascular pathology in the post-acute COVID-19 stage and regarding the impact of long COVID, which has been shown to be associated with potential infectious diseases associated with PVDs, primarily schistosomiasis, HIV, tuberculosis, pulmonary aspergillosis, *Wuchereria bancrofti*, *Clonorchis sinensis*, paracoccidioidomycosis, human herpesvirus 8, and scrub typhus. The likelihood of observing a high prevalence of PVDs due to co-infections in the post-COVID-19 era is pronounced. This could be explained by long COVID persistently impairing the physiological gas exchange caused by damaged lung architecture, including vasculature, hypoxic vasoconstriction, thrombotic events, direct viral damage, pro-inflammatory cytokines, and microthrombi [11].

PVDs are caused by a combination of genetic and environmental variables, as well as non-communicable and infectious diseases. Co-infection-related PVDs are so far poorly described, despite the associated global burden. The association of COVID-19 with other possible viral, bacterial, protozoal, and helminthic agents that may cause PVDs should be a cause for worry. According to one study, 19% of COVID-19 patients have co-infections, and 24% have superinfections [12]. The pooled prevalence of pathogen type stratified by co- or superinfection was 10% for viral co-infections and 4% for viral superinfections; 8% for bacterial co-infections and 20% for bacterial superinfections; and 4% for fungal co-infections and 8% for fungal superinfections [12]. Most of the potential co-infections inducing PVDs may be associated with COVID-19. Early estimates suggested that the prevalence of PH is estimated at 22% in patients with SARS-CoV-2 infection [13], and another study estimated that the incidence of PVDs in post-COVID-19 patients with suspected manifestations of PH is 70% [14]. The authors found that the prevalence of PVDs and right ventricular dysfunction in COVID-19 was 22% and 19%, respectively, and these were associated with increased hospital morbidity and mortality [15]. This prevalence may be higher in protracted COVID-19 coupled with PVD-related co-infections. Even though PVDs from noncommunicable diseases are more documented and reported, infectious diseases such as schistosomiasis, human immunodeficiency virus (HIV), pulmonary tuberculosis (PTB), filariasis, pulmonary aspergillosis, and others are significant sources of PVDs in their respective high-burden regions. PVDs associated with COVID-19 co-infections went understudied and underreported during the COVID-19 and post-COVID eras, and their impact on future PVDs is unknown. However, due to the pathophysiological effects of COVID-19/co-infections on pulmonary vascularization and their impacts in inducing PVDs, COVID-19 infections may be a genuine worry now and in the future.

Interestingly, of the co-infections that cause PVDs, it was reported that many HIV patients in Africa are also co-infected with schistosomiasis [16]. In the early stages, this study stipulated that both HIV proteins and *Schistosoma mansoni* egg-inducing granulomas are implicated in the release of inflammatory mediators known to cause adverse pulmonary arterial remodelling and endothelial cell injury [16]. Later, this hypothesis was confirmed by an experimental study which demonstrated that the combined pulmonary persistence of HIV proteins and schistosoma eggs, as it may occur in co-infected people, alters the cytokine landscape and targets the vascular endothelium for aggravated pulmonary vascular pathology [17]. This paper reviews the current literature on the potential link between COVID-19, parasitic co-infection, and PVDs. The authors acknowledge many points of the discussion remain hypothetical, reinforcing the urgent need of more research focused on infectious agents and the development of inflammatory PVDs. Despite the limitations, this review article is essential for the PVD field because it highlights a potential area of concern and encourages further research.

## 2. Methods

We did an electronic search using PubMed to find suitable publications, including COVID-19, associated with infectious diseases that cause PVDs, with an emphasis on case studies. The following search strategy was used: ((“Lung Diseases, Parasitic” [Majr]) AND (“COVID-19” [Mesh] OR “SARS-CoV-2” [Mesh] OR “Post-Acute COVID-19 Syndrome” [Mesh])) AND “Pulmonary Arterial Hypertension” [Mesh]). The search period was set from 2020 to 2024. After reading the abstracts, peer-reviewed articles were subgrouped into four categories: schistosomiasis and other helminthic diseases, HIV and viral infections such as other human herpesviruses, tuberculosis and other bacterial infections, and pulmonary aspergillosis and other fungal infections.

## 3. COVID-19 and Parasitic Co-Infection

### 3.1. Schistosomiasis and Other Helminthic Diseases

Over 230 million people (80% of whom are in Africa) are affected by schistosomiasis, which may cause PVDs in half of them [2,18] and the clinical presentation of pulmonary hypertension in 7–15% of them [2,19,20]. Schistosomiasis is endemic in more than 78 countries, including a high burden in all countries in sub-Saharan Africa, South East Asia, Brazil, Venezuela, and Yemen [21], and poses a serious disease burden to society, representing a major public health problem [22]. Approximately 10–20% of those with hepatosplenic disease, or 2–5 million people worldwide, develop PVDs, a progressive and fatal illness [20]. This high prevalence makes schistosomiasis one of the most common causes of PAH worldwide [19]. Initial symptoms include dyspnoea, dry cough, lower extremity edema, fatigue, and exercise intolerance [20]. As the disease progresses, patients can experience chest pain from right ventricular angina and syncope caused by depressed cardiac output, post-inflammatory cardiomyopathy, complex ventricular arrhythmias, and low systolic blood pressure [20,23]. Patients develop myocarditis, progressive right heart failure, and heart failure [23]. Physical examination may reveal a prominent pulmonic component, second heart sound (P2), right ventricular heave, and digital clubbing. Patients with frank right heart failure manifest cyanosis and peripheral edema or anasarca [20]. Radiographs may reveal cardiomegaly, particularly the dilatation of the right ventricle and right atrium, and enlarged pulmonary trunk and arteries, with pruning of the distal vasculature. Electrocardiography typically shows right ventricular hypertrophy or strain and right atrial enlargement and may also reveal a right bundle branch block [20]. Echocardiography demonstrates right ventricular dilatation, potentially compressing the left ventricle with septal bowing, usually accompanied by right atrial dilation, tricuspid valve regurgitation, and an increased pressure gradient across the tricuspid valve. Finally, right heart catheterization (when available) is used to confirm the diagnosis of PVDs in the absence of an elevated pulmonary artery occlusion pressure [20]. Other helminthic diseases can induce PVDs, such as *Wuchereria bancrofti*, a threadlike worm that causes filariasis (elephantiasis), and Clonorchis sinensis (Chinese liver fluke), which is a widespread parasite in southeast Asia, has been associated with cases of PVDs [2,24,25].

The literature review has revealed cases of helminthic diseases associated with COVID-19. Two cohort studies conducted in Ethiopia reported COVID-19/schistosomiasis co-infection rates of 4.5% [26] and 6.2% [27], respectively. COVID-19 infections and induced PVDs may potentialize multiple other co-infections. Among them, we found co-infections with *Wuchereria bancrofti*, a threadlike worm that causes filariasis (elephantiasis) [2,24,25]; *Clonorchis sinensis* (Chinese liver fluke), which is a widespread parasite in southeast Asia, has been associated with cases of pulmonary hypertension [28]. A case report of SARS-CoV-2/*Wuchereria bancrofti* co-infection was also reported [19].

The effect of COVID-19 on schistosomiasis-inducing PVD (COVID-Sch-PVD) is not well described. It is not clear if COVID-19 infection could trigger a host’s immune response to chronic indolent schistosomiasis infections or increase COVID-Sch-PVD morbidity and mortality in patients who are chronically exposed to the schistosoma parasite. Chronic portal hypertension can result in the opening of portocaval shunts, allowing schistosoma eggs to migrate from the portal system to the pulmonary tissue in which acute or long COVID may increase the likelihood of COVID-Sch-PVDs. Within the lungs, the eggs induce an immune response, which similarly results in severe granulomas that may be associated with severe inflammation related to COVID-19. This pathology of pulmonary arterial vascular remodelling could also be manifested, resulting in clinical COVID-Sch-PVD [29]. Severe COVID-19 has multisystem manifestations, including systemic thrombosis, cardiac injury, renal failure, and hepatic dysfunction. Following the acute phase, many patients with COVID-19 had symptoms due to persistent damage in several organs, including the lung vasculature and the sequelae of this damage, such as chronic lung fibrosis and, possibly, PVDs [30]. The patho-immunological interactions of both COVID-19 and helminthic diseases are complex. However, the acute COVID-19 phase, as well as long COVID, may interact in multiple pathways with chronic schistosomiasis and potentialize COVID-Sch-PVD (Figure 1).

Clinical studies on SARS-CoV-2 have revealed that the virus may cause alterations in pulmonary hemodynamics via processes comparable to those that cause PVDs, such as endothelial dysfunction, vascular leak, and thrombotic microangiopathy [8,9]. While most acute pulmonary embolisms and clots resolve with anticoagulation, clot persistence can lead to continued post-embolic symptoms of shortness of breath and the development of PVDs. About 30–50% of the patients have persistent defects up to 1 year after diagnosis [31,32]. The lumen can contain both fresh microthrombi and chronic thrombotic lesions with acute or chronic inflammatory cells and fibroblasts with variable degrees of organization. Increased PAH may be caused by a combination of pulmonary vasoconstriction, inward vascular wall remodelling, and in situ thrombosis, increasing vascular stiffness and narrowing the vascular lumen [33,34]. The endothelium in PVDs is thought to be activated or affected by chronic hypoxia, inflammation, viral infection, mechanical stretch, shear stress, and/or unknown causes [33]. This results the in altered production of endothelial mediators and the release of inflammatory mediators, which have been increasingly implicated in PVDs [33,35]. These inflammatory mediators, which include cell adhesion molecules, cytokines, chemokines, and growth factors, direct inflammatory cell recruitment and propagate a number of inflammatory pathways that lead to vascular cell proliferation, migration, and extracellular matrix deposition, all of which contribute to PH’s structural remodelling [33,36]. Bone morphogenic protein (BMP) and transforming growth factor-beta (TGF-β) synergistically activate regulatory T cells, hence diminishing inflammation and averting autoimmune diseases. A deficiency in bone morphogenetic protein receptor type 2 (BMPR2) function can lead to aberrant immune cell recruitment, heightened cytokine production, and inflammatory cell infiltration into the intima [11,37]. In the majority of patients (more than 80 percent), heritable PAH is caused by mutations in the BMPR2 gene that result in losses of function [38]. When severe cases of COVID-19 occur, there is a correlation between the loss of pericytes and cardiovascular inflammation [39]. Inflammatory processes are prominent in various forms of PVD. They are increasingly recognized as major pathogenic components of pulmonary vascular remodelling in PVDs related to more classical forms of inflammatory syndromes, fibrosis, and microvascular clotting, such as schistosomiasis (Figure 1).

Several regulatory pathways link helminths to Th2-mediated immune responses [40,41]. Enhanced vulnerability and a higher incidence of COVID-19 in schistosomiasis-endemic regions may be due to the downregulation of inflammation associated with the Th2 immune response in schistosomiasis infections in endemic countries (Figure 1). In Sch-PVD, the release of this cytokine is probably a consequence of the Th2 inflammation elicited by eggs deposited into lungs through a series of cellular and signalling events [42,43]. An interesting aspect is the fact that TGF-β activation seems to become autonomous and independent of the schistosome antigen, resulting in a persistent vascular disease despite parasite eradication [42]. Schistosoma infections are associated with a strong CD4 T-helper 2 (Th2) response [44]. The Th2 response follows in response to the egg and causes the production of a battery of cytokines such as interleukin (IL)-4, IL-6, IL-10 IL-13, IL-21, IL-31, acidic mammalian chitinase (AMCase), Ym1, and resistin-like molecule alpha1 (FIZZ1), as well as various chemokines [44,45]. The Th2 cells suppress the Th1 pro-inflammatory response and produce protective eosinophil-rich granulomatous lesions around newly deposited eggs, but they allow for the development of fibrosis [44,46]. The pathogenesis of severe disease in COVID-19 has been linked to the phenomenon of immune hyperactivation, which resembles that of a chronic inflammatory condition [27]. In the acute phase, SARS-CoV-2 infection drastically increases the production of pro-inflammatory cytokines, including IL2, IL7, interferon gamma-induced protein 10 (IP10), macrophage inflammatory protein-1 alpha (MIP1α), monocyte chemoattractant protein-1 (MCP1), and tumour necrosis factor alpha (TNFα), resulting in severe lung damage and fibrosis [47]. Furthermore, patients with long COVID-19 also showed dysregulated levels of matrix metalloproteinases (MMP)-2/MMP-9 [48], showing the substantial role of SARS-CoV-2 infection in lung fibrosis. Since PAH developing secondarily in pulmonary fibrosis patients (PF-PH) is a frequent co-morbidity, COVID-19/schistosomiasis-inducing PVD should be viewed as a serious threat. This could explain the rapid lung damage that could be observed in COVID-19/schistosoma co-infection [49] as both conditions may activate TGF-β, inducing an additive effect and consequently rapid lung fibrosis that could induce COVID-Sch-PVDs. However, this association may be underdiagnosed, particularly in endemic areas.

However, PVDs may be found in cases of SARS-CoV-2 associated with other helminthic co-infections in acute or long COVID periods due to the immunological interaction of both COVID-19 and helminthic PVDs complications associated with SARS-CoV-2/*Wuchereria bancrofti,* which may also be found in cases of *Clonorchis sinensis*/SARS-CoV-2 and hydatid cysts/SARS-CoV-2. Chronic helminthic infection suppresses both Th1 and Th2 responses by actively inducing the expansion of forkhead box protein 3 (FOXP^3+^) regulatory T cells, IL-10-producing B cells, and alternatively activated macrophages (AAMs), which together promote the release of regulatory cytokines such as TGF-β and IL-10 [50]. Because of a lack of antigen-specific T cells, helminthiasis is able to evade immune surveillance and persist in humans for extended periods of time [51]. The hallmarks include an upregulation of interleukin (IL)-10, a rise in the cytokine synthesis inhibitor, and a diminished T helper 1 response, while T helper 2 is inhibited. Phenotype absence in filariasis indicates T cell hyporesponsiveness, and the triggering of a CD4+ T cell response is usually necessary for the manifestation of the filariasis phenotype, which includes hydrocele, lymphedema, and elephantiasis. However, COVID-19 is characterized by several pro-inflammatory cytokine dysregulations, including type I interferon pathway activation, the increased secretion of IL-6, and the humoral immune pathway. It is believed that a severe strain of COVID-19 is exacerbated by this dysregulated immune response. Overall, COVID-19-associated chronic helminthic infections may be associated with an increased risk of PVDs. This could be explained by the pathophysiology of COVID-Sch-PVD, which could potentialize each other. Due to these potential pathophysiological mechanisms, PVDs may raise more than expected in affected COVID-19 areas, endemic schistosomiasis, and other helminthic diseases inducing PVDs. This could be explained by short- and long-term COVID-19 physiopathology in previous and future schistosomiasis cases.

### 3.2. HIV and Viral Infections Like Other Human Herpesviruses

The prevalence of PAH due to HIV infection is estimated at 8% (3.65–12.34%) [52]. The most significant burden of HIV lies in sub-Saharan Africa, where more than 20.9 million people were affected in 2022 [53]. Here, the prevalence of PAH due to HIV might be up to 0·5 per 1000 individuals, reaching 20–50 times higher than the prevalence of all PAH subtypes together in the developed world [54]. The incidence of PAH seems to be 1000 times higher in HIV-infected patients than in the general population [44]. Among 131 cases with HIV-PAH, a study found a progressive shortness of breath was the most common presenting symptom (85% of cases), followed by pedal edema (30%), non-productive cough (19%), fatigue (13%), syncope or near syncope (12%), and chest pains (7%) [55] associated with dyspnea and palpitations [14]. Right ventricular failure, myocarditis, progressive right heart failure, and heart failure may constitute the complications related to HIV-PAH. Experimental studies have shown that nicotinamide adenine dinucleotide phosphate (NADPH) oxidases are one of the leading players in the oxidative stress-mediated endothelial dysfunction on the dual hit of HIV-viral protein(s)-induced PAH [56] and that the presence of HIV-1 proteins likely impacts pulmonary vascular resistance and exacerbates hypoxia-induced PH [57] (Figure 2).

COVID-19 has been found to be associated with PVDs and pulmonary embolism [58]. The pooled prevalence of HIV among COVID-19 patients was 26.9% and was significantly higher in studies conducted in Africa compared to those conducted elsewhere [59]. Knowing that the two conditions may be associated with PVDs, it is unknown if they could interact or aggravate PVDs. A study showed that the average time interval between the diagnosis of HIV infection and the diagnosis of pulmonary hypertension was 33 months, while in 6% of cases, the diagnosis of HIV infection was established after the diagnosis of pulmonary hypertension [60]. Because of the probable effects of HIV and COVID-19 on the pathogenesis of PVDs, we estimated that this time would be shorter in cases of COVID-19 associated with HIV-inducing PVD (COVID-HIV-PVD). The pathobiology of COVID-19/HIV-inducing COVID-HIV-PVD may be the subject of more attention, particularly in high burden HIV countries where COVID-19 cases have been more prevalent. The duration of how long after SARS-CoV-2 infection this impaired HIV viral control might persist is not clear, but in conjunction with innate immune dysregulation, it is possible that SARS-CoV-2 infection might allow for HIV provirus to be reactivated and replicate, increasing the risk for developing COVID-HIV-PVD initiated by HIV viral and COVID-19 protein-mediated pulmonary vascular remodelling [11]. Coincidentally, both HIV and COVID-19 are viruses whose membrane fusion protein gp120 and spike protein, as well as other HIV proteins such as Tat and Nef, may trigger cell signalling that may promote pulmonary vascular remodelling, in addition to predisposing infected individuals to developing COVID-HIV-PVDs [61] (Figure 2). In severe PVDs, macrophages, lymphocytes, and dendritic cells are important inflammatory cellular components in the perivasculature of HIV tissues [44]. Hence, HIV-induced chronic inflammation and immune hyperactivation may enrich the pro-inflammatory milieu implicated in HIV-PVD. Inflammatory cells (macrophages and T and B lymphocytes) have been detected in various parts of the remodelled small pulmonary arteries and in plexiform lesions in many forms of PVDs. Furthermore, elevated serum levels of pro-inflammatory cytokines IL-1 and IL-6, 10, chemokines, and various types of autoantibodies (anti-nuclear, anti-endothelial, anti-fibroblast, and anti–fibrillin-1) have been detected in severe PVDs associated with HIV [44,62] (Figure 2). In the same line, COVID-19-induced changes in cells of the lung vascular wall, including endothelial cells, pericytes, smooth muscle cells (SMCs), and fibroblasts, have been observed [30]. The histopathological picture shows vascular intimal and medial hyperplasia, endarteritis obliterans, and severe inflammatory reactions [16,63]. The typical plexogenic arteriopathy is present in >80% of PLWH [16,63]. Based on the acute pathophysiology of SARS-CoV-2 connected to PVDs, as well as the protracted COVID pathophysiology associated with HIV, PVDs may pose a major concern in PLWH.

Other viral infections, such as human herpesvirus-8, showed evidence of PVD [64]. Cool et al. found that 62% of cells within and around the plexiform lesions from the lung tissue of patients with various causes of PVDs showed evidence of infection with human herpesvirus 8 (HHV-8) [44,64]. Herpes virus infection may exacerbate other pulmonary fibrosis-associated pathologies, and this may be hypothesized as more likely in the case of COVID-19/herpesvirus co-infections. PVDs are common in pulmonary fibrosis, affecting up to 40% of patients and being associated with worse outcomes [65]. Gamma herpes viruses are vasculotropic, and there is some evidence of their direct involvement in PVDs [66,67]. Herpes virus, particularly Epstein–Barr virus (EBV), in the explant lungs of patients with pulmonary fibrosis undergoing transplantation was associated with vascular remodelling. There was increased TGFβ expression and arterial intimal thickening, compared with disease and healthy control lungs, along with higher mean pulmonary arterial pressure and worse clinical outcomes [67]. Patients with post-COVID manifestations presented with the reactivation of EBV in 42.6%, human herpesvirus-8 (HHV-6) in 25.0%, and EBV plus HHV-6 in 32.4% [68]. A case of COVID-19/co-infection was described with left ventricular thrombus (LVT) [69]. Even though we did not find a case of COVID-19/HHV-8 inducing PVDs, it is therefore imperative to assess the role of HHV-8 in PVDs in endemic areas to provide further information on the role of both COVID-19 and herpes viruses that can bring about more clarification on the pathogenesis and interactions of acute stage and long-term stages of both viruses on PVDs.

### 3.3. Tuberculosis and Other Bacterial Infections

It is estimated that 10.6 million people (95% UI: 9.9–11.4 million) had TB worldwide [70]. Eight countries accounted for more than two-thirds of global TB cases: India (27%), Indonesia (10%), China (7.1%), the Philippines (7.0%), Pakistan (5.7%), Nigeria (4.5%), Bangladesh (3.6%), and the Democratic Republic of the Congo (3.0%) [70]. PVDs were incidental, finding approximately 15% of them with a preceding history of pulmonary tuberculosis (PTB) without any other forthcoming cause for PVDs [71]. The mean PASP was 29 mmHg, and RAP was estimated in 99 subjects with values ≤ 5 mmHg in 57 (57%), 5–10 mmHg in 34 (34%), and >10 mmHg in 8 (8%). Probable PVD-post TB was observed in nine subjects, yielding a prevalence of 9% (95% CI: 4.4–16.7%) [72]. This entity of “tuberculosis-related PVDs” is not well recognized and has not been given any space in all the classifications of PVDs [73]. Recently, Patel et al. have described PVDs in six out of fifty (12%) cases of PVDs developed from TB [11]. An association was found between PVDs and the number of previous PTB episodes, with each additional episode of PTB increasing the odds of PH-post PTB 2.13-fold [72]. A systematic review estimated the prevalence of COVID-19/PTB at 3% [2–5%]. However, this prevalence may be lower due to the high case fatality found post-mortem in 25% [3–47%] compared to clinical PTB diagnostics [74]. In a study assessing PAH among 14 cases treated for PTB, the estimated pulmonary artery systolic pressure (PASP) of 51 to 80 mm/Hg was found in 9 patients (64.3%). In contrast, a PASP of 40 to 50 mm/Hg was found in four patients (28.6%), and one patient had a PASP higher than 80 mm/Hg after 9 years [75]. This finding suggests that PVD will more likely develop with more extensively destroyed lung tissues and progress to poorer outcomes regardless of the regional distribution of PTB [75]. PAH in patients with TB-destroyed lungs (TDLs) was associated with the severity of lung destruction and led to more frequent exacerbation than that of TDL without PAH [76]. Similarly, Ryu et al. reviewed the clinical outcomes of 169 patients with TDL and reported that more extensive lung destruction was revealed as a risk factor for a poorer prognosis [76]. In fact, PTB can result in TDL, which is caused when parenchyma is damaged due to excessive M1 activity. Furthermore, macrophages unleash reactive oxygen (ROS), mount oxidative stress, and cause bystander damage to the surrounding tissue [77,78]. Other molecules, such as cathelicidins, defensins, cathepsins, matrix metalloproteases, and S100 proteins, drive connective tissue damage, fibrosis, and angiogenesis [77]. During the process of granuloma formation, activated alveolar macrophages (AMs) (realizing specific cytokines such as TGF-β and IL-10) invade the subtending epithelium and attract mononuclear cells from neighbouring blood vessels through chemotaxis, forming the cellular matrix of the early granuloma [79]. As a result, PTB may trigger PVDs. Some bacterial infections, such as Bordetella pertussis, may also trigger PVDs through their toxins. The mechanism of the development of PVDs is thought to result from residual pulmonary structural damage and pulmonary function abnormalities, leading to gas exchange abnormalities and chronic hypoxia [75,80]. In the literature review, case studies have shown the association between pulmonary embolism, TB, and COVID-19 [81,82].

The results showed that 50% of COVID-19 patients were co-infected or carried bacterial pathogens [83]. A global meta-analysis estimated the prevalence of COVID-19/PTB at 7.1% (95%CI: 4.0~10.8%) [84]. However, the incidence of active PTB among COVID-19 patients was higher in high tuberculosis burden countries. The Bordetella pertussis infection rate was significantly higher in COVID-19-positive patients [83]. PVDs associated with B. pertussis infection in infants are strongly correlated with disease severity, with 75% of infants that succumb to infection displaying features of pulmonary hypertension, compared with just 6% of those that survive infection [85]. Other plausible COVID-19/bacterial co-infections that could be associated with PVDs are numerous. In fact, COVID-19, and *Pneumocystis jirovecii* pulmonary co-infections were associated with extensive vascular thromboses in the pulmonary venous territory and a pulmonary edema focus below the subpleural hemorrhage area. Peri bronchial pulmonary artery thromboses were also observed [86]. Scrub typhus/COVID-19 co-infections were associated with a transient increase in the risk of vascular events, including pulmonary embolism [87].

TDL’s possible association with COVID-19 as a reactivation of latent TB in the setting of COVID-19 infection is plausible, given that the two diseases augment each other with a transient decrease in cellular immunity [74,88]. Furthermore, previous or active PTB was a risk factor for COVID-19 both in terms of severity and mortality, irrespective of HIV status [89]. In fact, the TGF-β activation pathways in both SARS-CoV-2 and PTB contribute to the production of fibrin, collagen, and secreted proteases (matrix metalloproteinases) associated with human cavities involved in the formation of fibrosis and tissue remodelling [74,90]. In addition to TGF- β and angiotensin-converting enzyme 2 (ACE2), other pathways can contribute to SARS-CoV-2-mediated lung fibrosis. In the same line, MCP-1 is a chemokine that causes lung fibrosis [74]. Studies reported that pulmonary thromboembolism (PTE) has been associated with TB [91]. As current or previous TB was associated with severe COVID-19, this can cause the development of COVID-19-associated coagulopathy, with features of both disseminated intravascular coagulation and thrombotic microangiopathy and PTE [92]. The alveolar and endothelial damage of smaller vessels may be followed by microvascular pulmonary thrombosis, which could then extend to larger vessels. As demonstrated in studies of COVID-19/TB co-infection producing pulmonary embolism, this may increase the risk of PTE [81,91]. In addition, there are permanent changes in lung architecture after TB, as shown in the TDL pathophysiology described above. These changes may co-exist with long COVID, and PTB and COVID-19 may also induce permanent fibrotic changes. Based on previous and current studies, PTB associated with COVID-19 may cause PVDs. In the same line, Bordetella pertussis infection as a consequence of PVD and the association of Bordetella pertussis with COVID-19 are well established. How Bordetella pertussis may induce PVDs remains unknown. In certain reports, hyperleukocyte thrombi have been found in small pulmonary blood arteries. It has also been hypothesized that vascular blockages brought on by leucocytosis brought on by pertussis toxin lead to an increase in pulmonary arterial pressure and the formation of PVDs [93,94]. Acute pulmonary vasoconstriction, hypoxia, microcirculation abnormalities, and clotting failure are other complications that pertussis pneumonia can cause [95]. Previous or present Bordetella pertussis infection in conjunction with severe COVID-19 may precipitate the development of PVDs since the pathophysiological mechanisms of both conditions may exacerbate each other.

We estimated that the impact of COVID-19 on TB-inducing PVDs (COVID-TB-PVD) may be more severe due to the interactions of these conditions in acute phases and long-term stages inducing vasculature, hypoxic vasoconstriction, thrombotic events, pro-inflammatory cytokines, microthrombi, and pulmonary fibrosis. However, COVID-19- or long COVID-induced COVID-TB-PVD should be the subject of investigations to improve its epidemiology and clinical profiles. Figure 3 depicts a hypothetical pathophysiology for acute SARS-CoV-2 and long COVID coupled with PTB and other bacterial infections that cause PVDs. Previous PTB history or GeneXpert associated with D-dimer echocardiograms are suggested in cases of COVID-TB-PVD.

### 3.4. Pulmonary Aspergillosis and Other Fungal Infections

Aspergillus is a fungus that can cause a wide variety of pulmonary disorders, ranging from non-invasive to invasive infections. As a result of its angiotrophism, it can produce microthrombus and pulmonary embolism, which can lead to an imbalance in the ventilation/perfusion ratio, which in turn makes hypoxia worse. Chronic pulmonary aspergillosis (CPA), like the majority of chronic granulomatous diseases, has the potential to raise the risk of PVDs. As with most chronic granulomatous diseases, chronic pulmonary aspergillosis (CPA) may increase the risk of PVDs. However, the association between pulmonary aspergillosis and PVDs is not well established.

COVID-19-associated pulmonary aspergillosis (CAPA) is considered a potentially life-threatening secondary infection in a large number of critically ill COVID-19 patients. Presumed CAPA may be present in as much as 19% of the intensive care unit (ICU) patients [96,97]. Studies have shown that CAPA has a high risk of pulmonary embolism [97,98]. The CAPA-related cumulative incidence of venous thromboembolism reported was 49% in patients admitted to the ICU [8]. Further, cases of extensive pulmonary fibrosis and chronic respiratory failure were reported in CAPA cases [99,100]. Even though the literature did establish the association between CAPA and PVDs, we hypothesize that CAPA may trigger PVDs, as the pathophysiological mechanisms have shown extended pulmonary fibrosis and thromboembolism as well as fungal angio-trophism, thereby facilitating SARS-CoV-2 entry into the target cell via ACE2. Endothelial cells express the receptors required for SARS-CoV-2 entry into the cells, such as ACE2 and transmembrane protease serine subtype 2 (TMPRSS2), which is known to be associated with endothelial injuries that may potentialize Aspergillus angio-tropism, which may then aggravate the SARS-CoV-2 infection. This could be one of the plausible explanations for the high CAPA incidence and severity in the ICU. Lastly, the early inflammatory hyperactivation pathway induced by the SARS-CoV-2 infection may be a major factor in establishing a highly permissive inflammatory environment that favours fungal pathogenesis.

In addition, fungal infections, like paracoccidioidomycosis, can cause PVDs in patients. A case reporting PTB, COVID-19, and *Paragonimus westermani* co-infection was also reported [101]. It is unknown how paracoccidioidomycosis associated with COVID-19 may trigger PVDs. Knowing that its pathophysiology may be more similar to CAPA-induced PVDs, as stated above, SARS-CoV-2 infection may be a primary component in creating a highly permissive inflammatory state that promotes fungal pathogenesis. Figure 4 summarizes different hypothetical mechanisms associated with COVID-19/co-infections inducing PVDs.

### 3.5. Gaps in Evidence

Based on the mechanisms linked with COV-19/co-infections inducing PVDs, there are numerous research gaps in COVID-Sch-PVD, COVID-HIV-PVD, COVID-TB-PVD, and CAPA-induced PVD. Table 1 summarizes research gaps, mechanisms, hypotheses, and future views on COVID-19/co-infections that cause PVDs.

## 4. A Multidisciplinary Diagnostic and Management Approach of PVDs Related to COVID-19 Co-Infections Is Important

The interactions between possible COVID-19 co-infections that cause PVDs are a novel subject that need additional research to gain a better understanding of the epidemiology, clinical profile, diagnostics, management, and prognostics. This multidisciplinary approach includes schistosomiasis, HIV, PTB, pulmonary aspergillosis, *Wuchereria bancrofti*, *Clonorchis sinensis*, paracoccidioidomycosis, and other infections that occur before or after acute COVID-19 or during long COVID phases, human herpesvirus 8, and scrub typhus. This review suggests that the diagnostics of the leading co-infections inducing PVDs should be undertaken in the context of COVID-Sch-PVD, COVID-HIV-PVD, and COVID-TB-PVD, or other related IDs in high-burden or endemic regions (Figure 4). Emphasis should be placed on COVID-Sch-PVD, COVID-HIV-PVD, and COVID-TB-PVD, as studies have shown that active and previous TB as well as HIV were found to be COVID-19 risk factors [89]. Furthermore, COVID-19 outcomes were assessed in relation to schistosomiasis prevalence and treatment coverage. COVID-19 outcomes, particularly active cases and recovery rates, were dramatically improved in schistosomiasis-free African countries [40].

In the case of the suspicion of COVID-Sch-PVD, COVID-HIV-PVD, COVID-TB-PVD, or other co-infections, the effects of COVID-19 and long COVID-19 on coagulation, inflammatory, and lung fibrosis biomarkers should be measured. Furthermore, cytological, histological, and medicinal imaging should be highlighted in cases of such co-infections. The American Thoracic Society’s PVD diagnosis guidelines, which are tailored for etiologic suspected infectious diseases, could be used to diagnose COVID-Sch-PVD, COVID-HIV-PVD, COVID-TB-PVD, and other co-infections. These assessments are critical since the epidemiology, clinical profile, diagnostics, treatment, and prognostics are all unknown. As a result, patients with a history of COVID-19 and high levels of D-dimer should undergo CT angiography in settings with a high burden of schistosomiasis, HIV, PTB, pulmonary aspergillosis, *Wuchereria bancrofti*, *Clonorchis sinensis*, paracoccidioidomycosis, human herpesvirus 8, and scrub typhus. Elevated D-dimer has been observed in patients with COVID-19 and long COVID. It is well known that elevated D-dimer levels are linked to acute pulmonary emboli and inflammatory disorders. D-dimer could be regarded as a simple, reliable, and low-cost diagnostic for tracking and following patients who have recovered from COVID-19. In the presence of co-infections, COVID-19 or long COVID may produce inflammation, which leads to endothelial function and coagulation abnormalities. In situations where COVID-Sch-PVD, COVID-HIV-PVD, COVID-TB-PVD, or other co-infections are suspected, D-dimer with echocardiography should be used in screening for PVD (Figure 5). Other tests include HIV Elisa, TB GeneXpert, herpes simplex serology, and the immunochromatographic test (ICT) for helminthiasis. Lastly, risk factors that may interact between PVDs, COVID-19, and infectious diseases associated with PVDs should be investigated, including advanced age, diabetes mellitus, living in close proximity to freshwater bodies, lower socio-economic level, poor sanitation, etc.

## 5. Conclusions

A wide range of infectious diseases can contribute to the development of PVDs during the acute COVID-19 and protracted COVID phases, increasing the global PVD burden. Based on previous research and pathophysiological pathways, this review identified COVID-19 and long COVID associated with parasites inducing PVDs as potent risks of increasing the burden of PVDs. Parasites inducing PVDs such as schistosomiasis, HIV, PTB, pulmonary aspergillosis, *Wuchereria bancrofti*, *Clonorchis sinensis*, paracoccidioidomycosis, human herpesvirus 8, and scrub typhus were found among COVID-19 and long COVID. However, it is hypothetical that this association may increase the risk of PVDs. Knowing that these potential associations could increase the PVD, PVD systematic screening is needed in the post-COVID era. Additionally, the epidemiology, clinical profiles, and prognostics of COVID-Sch-PVD, COVID-HIV-PVD, COVID-TB-PVD, and others remain unknown. Screening for COVID-Sch-PVD, COVID-HIV-PVD, COVID-TB-PVD, and other co-infections should be performed in suspected patients to better the epidemiology, clinical profiles, and prognostics of these co-infections. More experimental investigations with small animal, large animal, and in vitro models simulating COVID-Sch-PVD, COVID-HIV-PVD, and COVID-TB-PVD are required. This will enhance data for a better understanding of the relationship between PVD and various presumed infectious causes.

## Figures and Tables

**Figure 1 idr-17-00019-f001:**
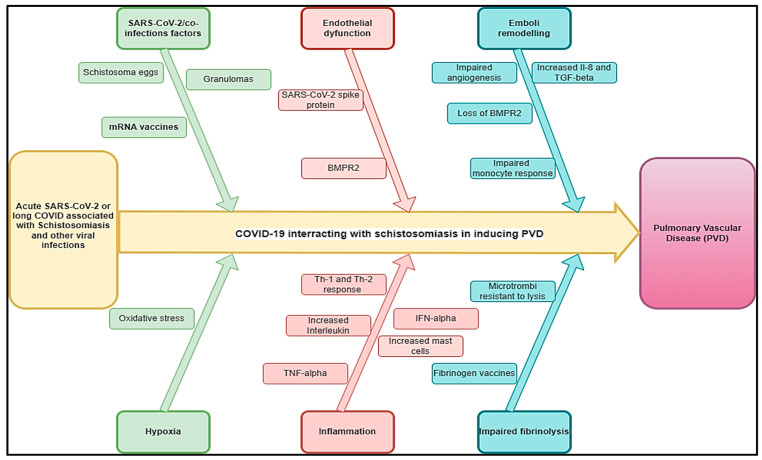
Hypothetical pathophysiology explaining acute SARS-CoV-2 and long COVID associated with schistosomiasis and other helminthic diseases inducing PVDs.

**Figure 2 idr-17-00019-f002:**
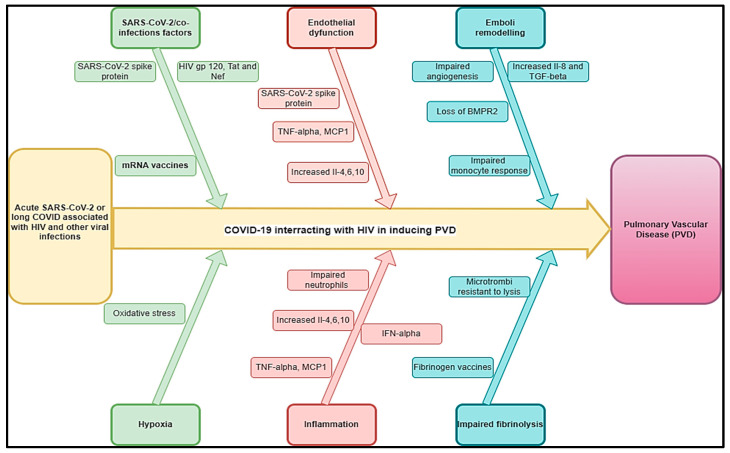
Hypothetical pathophysiology explaining acute SARS-CoV-2 and long COVID associated with HIV and other viral diseases inducing PVDs.

**Figure 3 idr-17-00019-f003:**
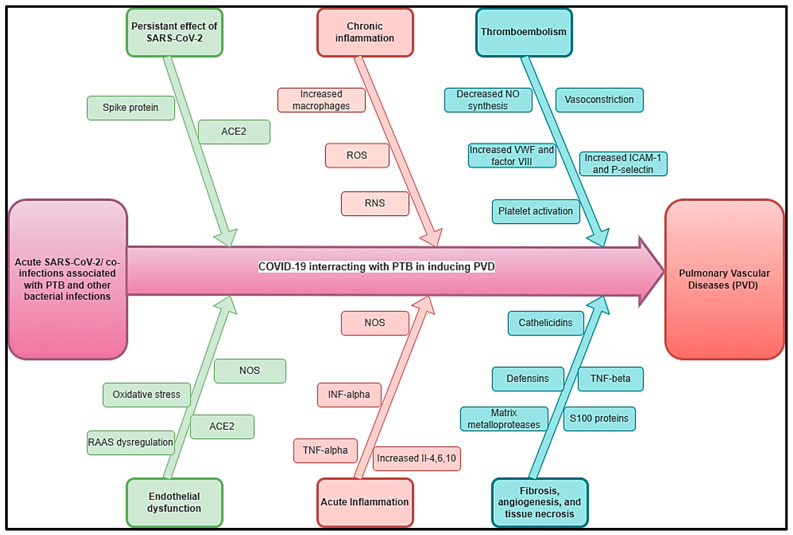
Hypothetical pathophysiology explaining acute SARS-CoV-2 and long COVID associated with PTB and other bacterial diseases inducing PVDs.

**Figure 4 idr-17-00019-f004:**
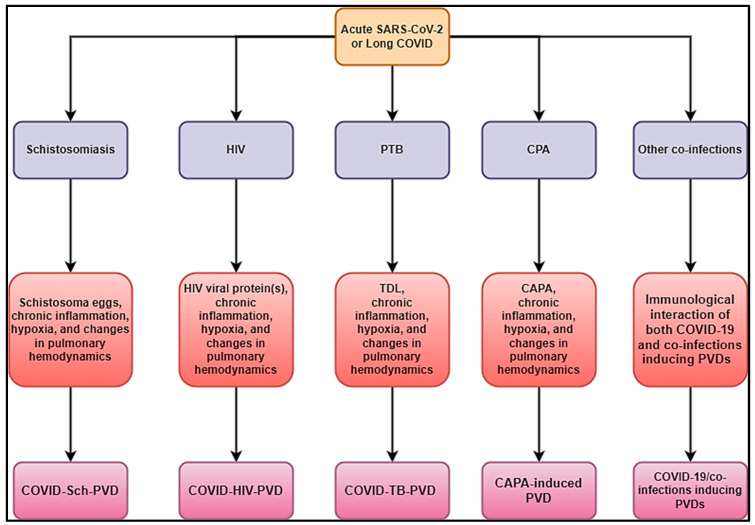
Hypothetical mechanisms associated with COVID-19/co-infections inducing PVDs.

**Figure 5 idr-17-00019-f005:**
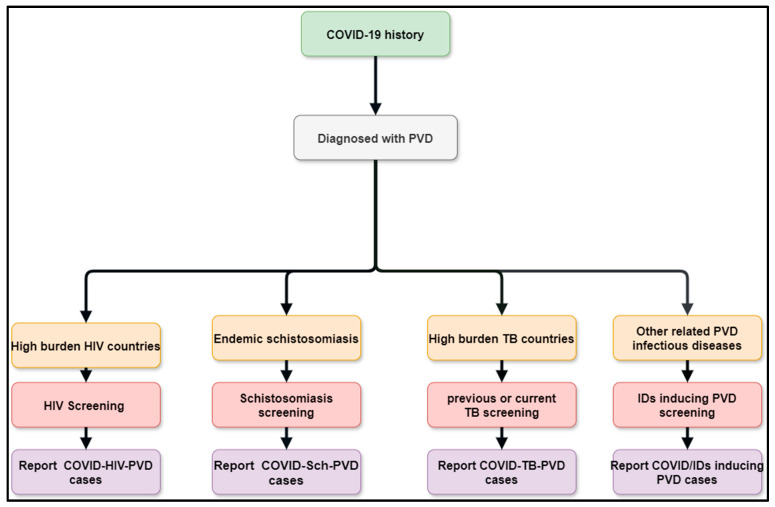
Flow diagram showing the methods of improving post COVID/co-infections inducing PVDs.

**Table 1 idr-17-00019-t001:** Research gaps, mechanisms, hypothesis, and future perspectives on COVID-19 associated with parasitic co-infection inducing PVDs.

Research Gaps	Plausible Mechanisms Inducing PVDs	Hypothesis	Future Perspectives
COVID-19 on schistosomiasis-inducing PVD (COVID-Sch-PVD) cases and mechanisms	Schistosoma eggs, chronic inflammation and immunological hyperactivation, chronic hypoxia, and changes in pulmonary hemodynamics, such as endothelial dysfunction, vascular leak, and thrombotic microangiopathy, are all associated with acute or long-term COVID.	How severe could COVID-Sch-PVD be compared to Sch-PVD? How could the clinical picture of COVID-Sch-PVD found in people with a history of Sch-PVD or COVID-Sch-PVD be found in the long COVID phase? What could be the most plausible pathophysiological pathways of COVID-Sch-PVD be?	Active clinical case search and subsequent investigation in high-burden COVID-19 and schistosomiasis settings in PVD cases. More experimental studies using small animal, large animal, and in vitro models.
COVID-19 associated with HIV inducing PVD (COVID-HIV-PVD) cases and mechanisms	HIV-viral protein(s), chronic inflammation and immune hyperactivation, chronic hypoxia, and alterations in pulmonary hemodynamics, including endothelial dysfunction, vascular leak, and thrombotic micro-angiopathy due to acute or long COVID.	How severe could COVID-HIV-PVD be compared to HIV-PVD? How could the clinical picture of COVID-HIV-PVD found in people with a history of HIV-PVD or COVID-HIV-PVD be found in the long COVID phase? What could be the most plausible pathophysiological pathways of COVID-HIV-PVD be?	Active clinical case search and subsequent investigation in high-burden COVID-19 and HIV settings in PVD cases. More experimental studies using small animal, large animal, and in vitro models.
COVID-19 on TB inducing PVDs (COVID-TB-PVD cases and mechanisms	TB-destroyed lung (TDL), chronic inflammation and immunological hyperactivation, chronic hypoxia, and changes in pulmonary hemodynamics, such as endothelial dysfunction, vascular leak, and thrombotic microangiopathy, as a result of acute or prolonged COVID.	How severe is COVID-TB-PVD in comparison to TB-PVD? How might the clinical picture of COVID-TB-PVD be found in people with a history of TB-PVD, or could it be found throughout the extended COVID phase? What are the most likely pathophysiological routes of COVID-TB-PVD?	Active clinical case search and subsequent investigation in high-burden COVID-19 and TB settings in PVD cases. More experimental studies using small animal, large animal, and in vitro models.
CAPA-induced PVDs cases and mechanisms	Chronic pulmonary aspergillosis (CPA), chronic inflammation and immune hyperactivation, chronic hypoxia, and alterations in pulmonary hemodynamics, including endothelial dysfunction, vascular leak, and thrombotic micro-angiopathy due to acute or long COVID.	How severe is CAPA-induc ing PVD in comparison to CPA-inducing PVD? What are the most likely pathophysiological routes of CAPA-inducing PVD?	Active clinical case search and subsequent study in high-burden filariasis and COVID-19 situations in PVD cases. More experimental research with small and big animal models, as well as in vitro models.

## Data Availability

Not applicable.

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
