# Peer review of "COVID-19 and Parasitic Co-Infection: A Hypothetical Link to Pulmonary Vascular Disease"

_2036-7449, 2025, doi:10.3390/idr17020019_

Round 1

Reviewer 1 Report

Comments and Suggestions for Authors

The authors of this manuscript present a literature review on the potential pathophysiological mechanisms in patients with COVID-19 and certain parasitic coinfections that might contribute to the development of pulmonary vascular disease (PVD). This manuscript addresses a clinically relevant and topical subject. PVD is relatively common in many populations worldwide, with highly diverse etiology and pathogenesis. It may result from a complex interplay of genetic and environmental factors, alongside both non-communicable and infectious diseases. Although common, PVDs associated with coinfections remain poorly understood. This knowledge gap forms the basis for the authors' literature review.

The manuscript is generally well-structured, with straightforward language and illustrative figures that effectively complement the content.

I have the following comments and recommendations for the authors:

  1. The Abstract should be more concise.
  2. The Introduction would benefit from transitional sentences that better link the topics discussed, such as pulmonary vascular disease, pulmonary vascular resistance, and chronic thromboembolic pulmonary hypertension (CTEPH). Currently, the initial sentences appear somewhat disconnected.
  3. Some sentences lack citation indexes, and these should be included.
  4. The manuscript's English language requires improvement to enhance readability and clarity.
  5. Consider concluding the manuscript with a section on "Gaps in Evidence" to highlight unresolved issues and areas for future research.
Comments on the Quality of English Language

 English language requires improvement to enhance readability and clarity.

Author Response

Please check the file in the attachment.

Reviewer 2 Report

Comments and Suggestions for Authors

In this review, the authors report on the prevalence of COVID-19, specifically pulmonary arterial hypertension (PAH) before and after COVID-19. The prevalence of this disease, which the authors say was previously estimated to be about 1 per 100,000, has increased due to risk factors (obesity, cardiovascular disease, and some infections) for pulmonary vascular disease (PVD) associated with COVID-19 and its complications. It is of interest whether this cause is related to COVID-19. The authors focus their review on this point and suggest that it will provide useful information for various risk managers in clinical practice and in society.

On the other hand, many of the possibilities that the basis for the increased prevalence associated with COVID-19 is caused by a combination of various pathogens are not hypothetical. We have some recommendations to substantiate this hypothesis, but please take this into account to deepen the discussion below.

1. Many of the findings on the impact of COVID-19 and parasite co-infection on PVD mentioned in this review are based on hypotheses and lack solid clinical data and evidence. We consider this to be particularly problematic, as many of the conclusions are speculative.

2. The epidemiological data on PVD and COVID-19/parasite co-infection in this study may be biased due to regional characteristics and lack of reporting. Is it possible to utilize data from other regions or other parts of the world?

3. Although this review presents risks and potential pathophysiological mechanisms, we feel that there are insufficient suggestions for specific treatment strategies and interventions. Please add any practical guidelines for clinical application.

4. With regard to the interaction between COVID-19 and parasitic infections, clarification of how individual infections and patient characteristics are related is required to clarify their causal relationship. Please include regional characteristics, race, age, gender, etc. to deepen recommendations for future development.

Minor point

1.     The quality of the diagrams is low. Text and arrows are difficult to read due to rough image quality (low dpi?). Please correct.

Author Response

(The authors gave the same response as above.)

Reviewer 3 Report

Comments and Suggestions for Authors

Manuscript entitled "COVID-19 and Parasitic Coinfection: A Hypothetical Link to Pulmonary Vascular Disease" by Peter S. Nyasulu, et al.

This manuscript explores the hypothetical link between COVID-19, parasitic co-infections, and the development of pulmonary vascular diseases (PVDs). It provides a comprehensive review of current literature and presents several mechanistic hypotheses. While the topic is highly relevant, given the evolving challenges of COVID-19 and its sequelae, several areas require improvement to enhance clarity, focus, and impact.

Comments:

  1. The introduction should be expanded to highlight the global burden of parasitic diseases and their established role in PVD pathogenesis.
  2. The manuscript discusses several potential mechanisms by which parasitic co-infections might exacerbate PVDs in the context of COVID-19. However, these mechanisms are described in a fragmented manner. A schematic figure summarizing the interplay of parasitic infections, COVID-19, and immune responses leading to PVD would significantly improve readability.
  3. Consider including a table summarizing the key characteristics of each co-infection (e.g., schistosomiasis, HIV, TB) and their contribution to PVDs.
  4. Why might certain parasitic co-infections (e.g., schistosomiasis) be more likely to interact with COVID-19 compared to others?
  5. Are there regional differences in the prevalence or impact of these co-infections that should be highlighted?
  6. Expand on potential knowledge gaps.
  7. The manuscript briefly mentions the need for further research but should provide more specific recommendations
  8. The methodology for the literature review is unclear. Include details on Search strategy and Criteria for selecting specific co-infections for discussion.

Author Response

(The authors gave the same response as above.)

Round 2

Reviewer 2 Report

Comments and Suggestions for Authors

The authors have made appropriate revisions and additions to my previous comments. I hope that the paper will be published. I do not require any further revisions.

Reviewer 3 Report

Comments and Suggestions for Authors

The authors have adequately addressed my comments, and the manuscript can be accepted for publication.